# DeVRF: Fast Deformable Voxel Radiance Fields for Dynamic Scenes

**Jia-Wei Liu**[1][*]**, Yan-Pei Cao**[2]**, Weijia Mao**[1]**, Wenqiao Zhang**[4]**, David Junhao Zhang**[1]**,
Jussi Keppo**[5,6]**, Ying Shan**[2]**, Xiaohu Qie**[3]**, Mike Zheng Shou**[1][†]

[1] Show Lab, National University of Singapore  [2] ARC Lab, [3] Tencent PCG
[4] National University of Singapore  [5] Business School, National University of Singapore
[6] Institute of Operations Research and Analytics, National University of Singapore

## Abstract

Modeling dynamic scenes is important for many applications such as virtual reality and telepresence. Despite achieving unprecedented fidelity for novel view synthesis in dynamic scenes, existing methods based on Neural Radiance Fields (NeRF) suffer from slow convergence (*i.e.*, model training time measured in *days*). In this paper, we present DeVRF, a novel representation to accelerate learning dynamic radiance fields. The core of DeVRF is to model both the 3D canonical space and 4D deformation field of a dynamic, non-rigid scene with explicit and discrete voxel-based representations. However, it is quite challenging to train such a representation which has a large number of model parameters, often resulting in overfitting issues. To overcome this challenge, we devise a novel *static* $\rightarrow$ *dynamic* learning paradigm together with a new data capture setup that is convenient to deploy in practice. This paradigm unlocks efficient learning of deformable radiance fields via utilizing the 3D volumetric canonical space learnt from multi-view static images to ease the learning of 4D voxel deformation field with only few-view dynamic sequences. To further improve the efficiency of our DeVRF and its synthesized novel view's quality, we conduct thorough explorations and identify a set of strategies. We evaluate DeVRF on both synthetic and real-world dynamic scenes with different types of deformation. Experiments demonstrate that DeVRF achieves two orders of magnitude speedup (**100× faster**) with on-par high-fidelity results compared to the previous state-of-the-art approaches. The code and dataset are released in `https://github.com/showlab/DeVRF`.

## 1  Introduction

Free-viewpoint photorealistic view synthesis techniques from a set of captured images unleash new opportunities for immersive applications such as virtual reality, telepresence, and 3D animation production. Recent advances in this domain mainly focus on static scenes, e.g., Neural Radiance Fields (NeRF) [20], which implicitly represent rigid static scenes using 5D (spatial locations $(x, y, z)$ and view directions $(\theta, \varphi)$) neural radiance fields. Although achieving unprecedented fidelity for novel view synthesis, NeRF were mainly exploited under static scenes. To unlock dynamic view synthesis, existing NeRF-based approaches either learn an additional MLP-based deformation field that maps coordinates in dynamic fields to NeRF-based canonical spaces [27, 24, 25, 38] or model dynamic scenes as 4D spatio-temporal radiance fields with relatively large MLPs [16, 7].

---

[*]Work is partially done during internship at ARC Lab, Tencent PCG.
[†]Corresponding Author.

36th Conference on Neural Information Processing Systems (NeurIPS 2022).

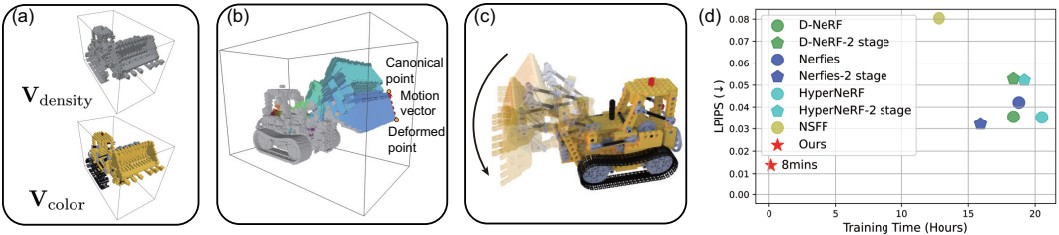

Figure 1: The 3D canonical space **(a)** and the 4D deformation field **(b)** of DeVRF for neural modeling of a non-rigid scene **(c)**. **(d)**: The comparison between DeVRF and SOTA approaches.

Despite being promising, NeRF is notoriously known for suffering from lengthy optimization time. This issue becomes particularly prominent for non-rigid, dynamic scenes because the aforementioned NeRF-based approaches require extra computation for the deformation MLPs [27, 24, 25, 38] or time-varying texture/density querying [16, 7], resulting in quite long training time (in "days").

This motivates us to improve the learning efficiency of dynamic radiance fields. Recent advances in static NeRF [35, 45] show that employing voxel grids, such a volumetric representation, can achieve fast convergence. To adapt for dynamic scenes, one straightforward approach is to incorporate such a volumetric representation into the dynamic radiance field for fast neural modeling. In this paper, we present a novel deformable voxel radiance field (DeVRF) that models both the 3D canonical space and 4D deformation field of a non-rigid, dynamic scene with explicit and discrete voxel-based representations, as illustrated in Fig. 1 (a-c). However, we empirically observe that recklessly learning such a representation in dynamic radiance fields tends to plunge into the local optimum, *i.e.*, the overfitting issue, due to the large number of parameters in DeVRF.

To overcome this overfitting issue, we power our DeVRF with two novel designs: **(1)** We devise an efficient and practical learning paradigm, *i.e.*, **static → dynamic**, for learning deformable radiance fields. The key idea behind this is that the 3D volumetric canonical space learned from multi-view static images can introduce *inductive bias* [3] to unlock efficient learning of deformable radiance fields. Further, with such 3D priors, a dynamic scene can be effectively modeled with only a few fixed cameras. We argue that such a few-fixed-cameras setup for dynamic scene data capture is more convenient than the moving camera (such as the setup used in D-NeRF [27]) in practice. **(2)** Based on the *static → dynamic* paradigm, we conduct extensive explorations and identify a set of strategies customised for DeVRF to improve its efficiency and effectiveness. These include a coarse-to-fine training strategy for the 4D deformation field to further improve efficiency, and three objectives to encourage our DeVRF to reconstruct dynamic radiance fields with high fidelity: deformation cycle consistency, optical flow supervisions, and total variation regularization.

Fig. 1 (d) shows that on five inward-facing synthetic scenes, two forward-facing real-world scenes and one inward-facing real-world scene, our approach enables fast dynamic radiance field modeling in about 10 minutes on a single NVIDIA GeForce RTX3090 GPU. This is **100×** faster than SOTA approaches with comparable novel view synthesis quality.

To summarize, the major contributions of our paper are:

- A novel perspective of DeVRF is presented that enables fast non-rigid neural scene reconstruction, which achieves an impressive 100× speedup compared to SOTA approaches with on-par high-fidelity.
- To the best of our knowledge, we are the first to incorporate the 4D voxel deformation field into dynamic radiance fields.
- We devise a *static → dynamic* learning paradigm that can boost performances with a low-cost yet effective capture setup.

## 2  Related Work

**Novel View Synthesis for Static Scenes.** Earlier approaches [14, 31, 4, 6, 9, 21] tackle novel view synthesis by first building an explicit 3D reconstruction of a static scene, such as voxels and meshes, and then rendering novel views based on the reconstructed model. On the other hand, multi-plane

images [48, 19] represent a scene with multiple images at different depths and can reconstruct scenes with complex structures. Recently, NeRF [20] achieves unprecedented fidelity for novel view synthesis by modeling static scenes with neural radiance fields. Subsequent works have extended NeRF to different scenarios, such as few-view novel view synthesis [12], multi-scale representation [2], and larger scenes [36, 43, 29]. However, these methods mainly focus on static scenes while the dynamic radiance fields reconstruction is more practical.

**Novel View Synthesis for Dynamic Scenes.** In order to capture dynamic scenes with non-rigidly deforming objects, traditional non-rigid reconstruction approaches require depth information as additional input or only reconstruct sparse geometry [23, 11, 44, 5, 34]. VolumeDeform [11] introduces a unified volumetric representation to encode both the scene's geometry and its motion but requires RGB-D sequences as input. Neural Volumes [18] represents dynamic objects with a 3D voxel grid plus an implicit warp field, but requires an expensive multi-view capture rig and days to train. Recent studies have built upon NeRF [20] and extended it to dynamic neural radiance field reconstruction by learning a mapping from dynamic to canonical field [27, 24, 25, 38] or building a 4D spatio-temporal radiance field [42, 16, 7, 15]. D-NeRF [27] learns a deformation field that maps coordinates in a dynamic field to a NeRF-based canonical space. Nerfies [24] further associates latent codes in the deformation MLP and the canonical NeRF to tackle more challenging scenes such as moving humans. HyperNeRF [25] proposes to model the motion in a higher dimension space, representing the time-dependent radiance field by slicing through the hyperspace. In contrast, Video-NeRF [42] models the dynamic scene as 4D spatio-temporal radiance fields and addresses motion ambiguity using scene depth. Sharing a similar idea on the 4D spatio-temporal field, NSFF [16] represents a dynamic scene as a time-variant function of geometry and appearance, and warps dynamic scene with 3D scene motion. Lastly, several NeRF-based approaches have been proposed for modeling dynamic humans [8, 41, 17, 26, 32] but can not directly generalize to other scenes. Although achieving promising results, existing methods require days of GPU training time, which is undesirable in real-world applications.

**NeRF Acceleration.** In the light of NeRF's substantial computational requirements for training and rendering, recent papers have proposed methods to improve its efficiency. A line of work [46, 28, 10] focuses on NeRF rendering acceleration and has achieved encouraging results. As for training acceleration, DVGO [35] models the radiance field with explicit and discretized volume representations, reducing training time to *minutes*. Plenoxels [45] employs sparse voxel grids as the scene representation and uses spherical harmonics to model view-dependent appearance, reaching a similar training speedup. Finally, Instant-ngp [22] proposes multiresolution hash encoding; together with a highly optimized GPU implementation, it can produce competitive results after *seconds* of training. However, existing acceleration methods only focus on static scenes, while hardly any research, to our best knowledge, has studied NeRF acceleration for dynamic scenes. Very recently, Fourier PlenOctrees [39] extends PlenOctrees [46] to dynamic scenes by processing time-varying density and color in the frequency domain; however, the data capturing setup is expensive, and it still requires hours of training. Instead, our proposed algorithm, DeVRF, offers a superior training speed while only requires a few cameras for data capture.

## 3 Method

### 3.1 Capture Setup

Deformable scenes undergo various types of deformations and motions, which can result in different scene properties such as object poses, shapes, and occlusions. Therefore, capturing and modeling deformable scenes is nontrivial even for professional photographic studios. Existing approaches [39, 13, 49, 1] attempt to capture 360° inward-facing dynamic scenes with multi-view sequences and thus require dozens of high-quality cameras. On the other hand, D-NeRF [27] reconstructs deformable radiance fields from a sparse set of synthetic images rendered from a moving monocular camera. However, in practice, it is particularly challenging to capture real-world 360° inward-facing dynamic scenes with a single moving camera due to various types of deformations and resulting occlusions in dynamic scenes, especially for scenes undergoing fast deformations. As a result, subsequent studies [24, 25, 16, 7] only capture forward-facing videos of real-world dynamic scenes with a monocular camera.

Table 1: Comparisons of capture setups for dynamic scenes.

| Approach | No. of cameras | Cost | Supported real-world use cases |
|---|---|---|---|
| D-NeRF[27], Nerfies[24] | Monocular | Low | Forward-facing scenes, slow reconstruction in *days*. |
| Neural Volumes[18] | Multiple (34) | High | 360° inward-facing scenes, slow reconstruction in *days*. |
| Fourier PlenOctrees[39] | Multiple (60) | High | 360° inward-facing scenes, fast reconstruction in 2hrs. |
| Ours | Few (4) | Low | 360° inward-facing and forward-facing scenes, super-fast reconstruction in 10mins. |

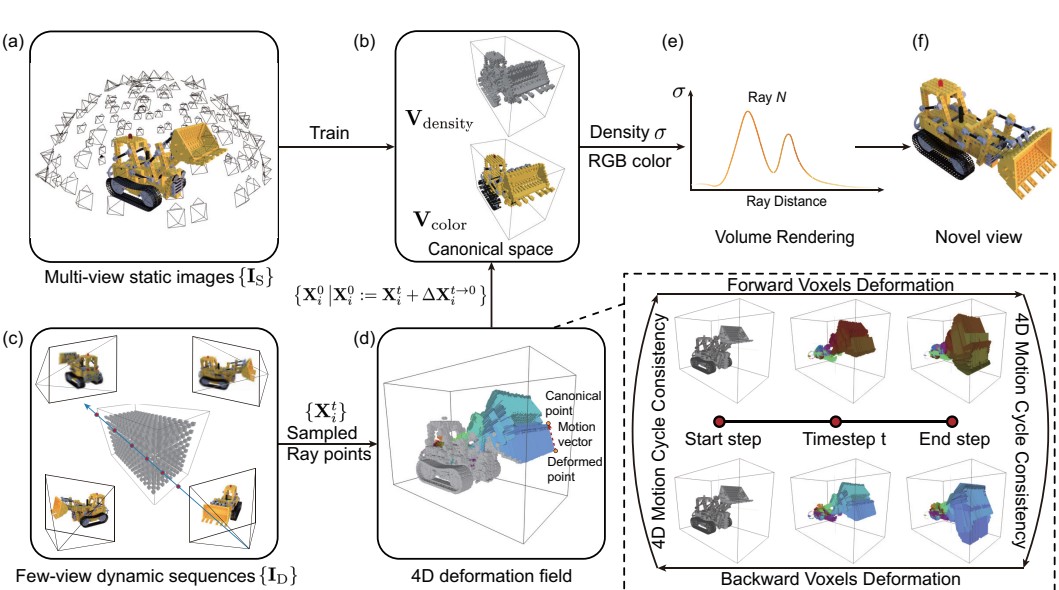

Figure 2: **Overview of our method.** In the first stage, DeVRF learns a 3D volumetric canonical prior **(b)** from multi-view static images **(a)**. In the second stage, a 4D deformation field **(d)** is jointly optimized from taking few-view dynamic sequences **(c)** and the 3D canonical prior **(b)**. For ray points sampled from a deformed frame, their deformation to canonical space can be efficiently queried from the 4D backward deformation field **(d)**. Therefore, the scene properties (*i.e.*, density, color) of these deformed points can be obtained through linear interpolation in the 3D volumetric canonical space, and novel views **(f)** can be accordingly synthesized by volume rendering **(e)** using these deformed sample points.

Compared to dynamic scenes, it is much easier in practice to do multi-view capture for real-world static scenes with a monocular moving camera. Therefore, we propose to separate the capture process of a dynamic scene into two stages: the first stage captures a static state using a moving monocular camera, and the second stage captures the scene in motion using a few fixed cameras. In this capture setup, the multi-view static images provide complete 3D geometry and appearance information of the scene, while few-view dynamic sequences show how the scene deforms in 3D space over time; the entire capture process only requires a few cameras. Tab. 1 compares our capture process with existing approaches in terms of the number of required cameras, cost, and supported real-world use cases.

### 3.2 Deformable Voxel Radiance Fields

As illustrated in Fig. 2, we present DeVRF to model both the 3D canonical space and 4D deformation field of a non-rigid scene with explicit and discrete voxel-based representations. This volumetric representation allows us to efficiently query the deformation, density, and color of any 3D point at any time step in a deformable scene, thus largely improving the training and rendering efficiency. In addition, we devise a *static → dynamic* learning paradigm that first learns a 3D volumetric canonical prior from multi-view static images (Fig. 2(a-b)) and transfers such prior to dynamic radiance fields reconstruction (Fig. 2(c-f)).

**3D Volumetric Canonical Space.** We take inspiration from the volumetric representation of DVGO [35] and model the scene properties such as density and color of our 3D canonical space into voxel

grids. Such representation enables us to efficiently query the scene property of any 3D point via trilinear interpolation of its neighboring voxels,

$$\text{Tri-Interp}\left([x, y, z], \mathbf{V}_p\right) : \left(\mathbb{R}^3, \mathbb{R}^{C \times N_x \times N_y \times N_z}\right) \rightarrow \mathbb{R}^C, \forall p \in \{\text{density, color}\} \tag{1}$$

where $C$ is the dimension of scene property $\mathbf{V}_p$. $N_x$, $N_y$, and $N_z$ are the voxel resolutions of $\mathbf{V}_p$ in $x$-, $y$-, $z$- dimension.

As shown in Fig. 2(a-b), we learn the 3D volumetric canonical prior, *i.e.*, density grid $\mathbf{V}_{\text{density}}$ and color grid $\mathbf{V}_{\text{color}}$, with multi-view static images $\{\mathbf{I}_S\}$ via volumetric rendering. Following DVGO [35], we employ softplus and post-activation after the trilinear interpolation of a 3D point in $\mathbf{V}_{\text{density}}$, as they are critical for sharp boundary and high-frequency geometry reconstruction. We also apply a shallow MLP after the trilinear interpolation of a 3D point in $\mathbf{V}_{\text{color}}$ to enable view-dependent color effects [35]. In our static $\rightarrow$ dynamic learning paradigm, the learned 3D volumetric canonical prior provides critical knowledge of the 3D geometry and appearance of the target dynamic scene, as few-view dynamic sequences alone struggle to reconstruct a complete deformable radiance field with high fidelity (as shown in Section 4).

**4D Voxel Deformation Field.** We employ a 4D voxel deformation field $\mathbf{V}_{\text{motion}}$ to efficiently represent the motion of a deformable scene. As shown in Fig. 2(d), the arrow directions represent the motions of voxels, the color denotes the motion direction, and the arrow magnitude denotes the motion scale. To synthesize a novel view at time step $t$, we shoot rays through image pixels and sample ray points $\mathcal{X}_t = \{\mathbf{X}_i^t\}$ in 3D space. The 3D motion $\Delta\mathcal{X}_{t \rightarrow 0} = \{\Delta\mathbf{X}_i^{t \rightarrow 0}\}$ from $\mathcal{X}_t$ to their corresponding 3D points in the canonical space $\mathcal{X}_0 = \{\mathbf{X}_i^0 \mid \mathbf{X}_i^0 = \mathbf{X}_i^t + \Delta\mathbf{X}_i^{t \rightarrow 0}\}$ can be efficiently queried through quadruple interpolation of their neighboring voxels at neighboring time steps in the 4D backward deformation field,

$$\text{Quad-Interp}\left([x, y, z, t], \mathbf{V}_{\text{motion}}\right) : \left(\mathbb{R}^4, \mathbb{R}^{N_t \times C \times N_x \times N_y \times N_z}\right) \rightarrow \mathbb{R}^C, \tag{2}$$

where $C$ is the degrees of freedom (DoFs) of the sample point motion. We use $C = 3$ in this paper, *i.e.*, assign a displacement vector to each sample point. $N_t$ is the number of key time steps that can be user-defined based on the scene motion properties.

Therefore, scene properties of $\mathcal{X}_t$ can then be obtained by querying the scene properties of their corresponding canonical points $\mathcal{X}_0$ through trilinear interpolation in the volumetric canonical space. Finally, pixel colors can be calculated through volume rendering with the sampled scene properties along each ray, as illustrated in Fig. 2(e-f).

### 3.3 Optimization

Training the DeVRF is quite challenging, mainly because a large number of model parameters may lead to overfitting or suboptimal solutions. This section describes the training strategy and optimization losses that we design to facilitate fast optimization of the DeVRF.

**Coarse-to-Fine Optimization.** For a dense 4D voxel deformation field with $N_t \times C \times N_x \times N_y \times N_z$ resolution, there could be millions of free parameters, which are prone to overfitting and suboptimal solutions. To solve this problem, we employ a coarse-to-fine training strategy. Specifically, in our experiments, we progressively up-scale the $x$-$y$-$z$ resolution of the 4D voxel deformation field from $10 \times 10 \times 10$ to $160 \times 160 \times 160$. With this strategy, the 4D voxel deformation field first learns a rough motion at the coarse stage, which is thereafter progressively refined in finer stages. Our experiments demonstrate that the coarse-to-fine strategy can effectively smoothen the optimization landscape of the 4D voxel deformation field and remove most suboptimal solutions, thus largely improving the training efficiency and accuracy.

**Re-rendering Loss.** With sampled properties at $\mathcal{X}_t$, the color of a pixel can be calculated through volume rendering, *i.e.*, by integrating the density and color of $\mathcal{X}_t$ along a ray $\mathbf{r}$ [20]:

$$\hat{C}(\mathbf{r}) = \sum_{i=1}^{N_r} T_i \left(1 - \exp\left(-\sigma_i \delta_i\right)\right) c_i + T_{N_r+1} \mathbf{c}_{\text{bg}}, \ T_i = \exp\left(-\sum_{j=1}^{i-1} \sigma_j \delta_j\right), \tag{3}$$

where $N_r$ is the number of sampled deformed points along the ray, $T_i$ represents the probability of light transmitting through ray $\mathbf{r}$ to the $i$-th sampled point, and $1 - \exp\left(-\sigma_i \delta_i\right)$ is the probability

that light terminates at the $i$-th point. $\delta_i$ is the distance between adjacent sampled points, and $\sigma_i$, $c_i$ denote the density and color of deformed point $i$, respectively. $\mathbf{c}_{\text{bg}}$ is the pre-defined background color.

Given the few-view training dynamic sequences with calibrated poses $\{\mathbf{I}_{\text{D}}\}$, DeVRF is optimized by minimizing the photometric MSE loss between the observed pixels color $C(\mathbf{r})$ and the rendered pixels color $\hat{C}(\mathbf{r})$ :

$$\mathcal{L}_{\text{Render}} = \frac{1}{|\mathcal{R}|} \sum_{\mathbf{r} \in \mathcal{R}} \left\| \hat{C}(\mathbf{r}) - C(\mathbf{r}) \right\|_2^2 , \tag{4}$$

where $\mathcal{R}$ is the set of rays in a mini-batch.

**4D Deformation Cycle Consistency.** As illustrated in Fig. 2(d), the forward deformation is modeled as 4D forward deformation voxels with the same resolution as the 4D backward deformation voxels, and we enforce 4D deformation cycle consistency between backward and forward motion, which regularizes the learned deformation field. In the 4D deformation cycle, backward motion vectors $\Delta \mathcal{X}_{t \to 0}$ models the motion from $\mathcal{X}_t$ to $\mathcal{X}_0$; in contrast, forward motion vectors $\Delta \mathcal{X}_{0 \to t}$ models the motion from $\mathcal{X}_0$ to their corresponding 3D points in the dynamic space $\tilde{\mathcal{X}}_t = \{\tilde{\mathbf{X}}_i^t \mid \tilde{\mathbf{X}}_i^t = \mathbf{X}_i^0 + \Delta \mathbf{X}_i^{0 \to t}\}$. The 4D motion cycle consistency can now be realized by minimizing the following cycle consistency loss $\mathcal{L}_{\text{Cycle}}(t)$,

$$\mathcal{L}_{\text{Cycle}}(t) = \frac{1}{2N_s} \sum_{i=1}^{N_s} \left\| \mathbf{X}_i^t - \tilde{\mathbf{X}}_i^t \right\|_2^2 , \tag{5}$$

where $N_s$ is the number of sampled 3D points in a mini-batch.

**Optical Flow Supervision.** The DeVRF is indirectly supervised by 2D optical flows estimated from consecutive frames of each dynamic sequence using a pre-trained RAFT model [37]. For $\mathcal{X}_t$ and their corresponding $\mathcal{X}_0$, we first compute the corresponding 3D points of $\mathcal{X}_0$ at $t-1$ time step via forward motion $\tilde{\mathcal{X}}_{t-1} = \{\tilde{\mathbf{X}}_i^{t-1} \mid \tilde{\mathbf{X}}_i^{t-1} = \mathbf{X}_i^0 + \Delta \mathbf{X}_i^{0 \to t-1}\}$. After that, we project $\tilde{\mathcal{X}}_{t-1}$ onto the reference camera and get their pixel locations $\tilde{\mathcal{P}}_{t-1} = \{\tilde{\mathbf{P}}_i^{t-1}\}$, and compute the induced optical flow with respect to the pixel location $\mathcal{P}_t = \{\mathbf{P}_i^t\}$ from which the rays of $\mathcal{X}_t$ are cast. We enforce the induced flow to be the same as the estimated flow by minimizing $\mathcal{L}_{\text{Flow}}(t)$,

$$\mathcal{L}_{\text{Flow}}(t) = \frac{1}{|\mathcal{R}|} \sum_{\mathbf{r} \in \mathcal{R}} \sum_{i=1}^{N_r} w_{\mathbf{r},i} \left| \left( \tilde{\mathbf{P}}_{\mathbf{r},i}^{t-1} - \mathbf{P}_{\mathbf{r},i}^t \right) - \mathbf{f}_{\mathbf{P}_{\mathbf{r},i}^t} \right| , \tag{6}$$

where $w_{\mathbf{r},i} = T_i \left(1 - \exp\left(-\sigma_i \delta_i\right)\right)$ is the ray termination weights from Eq. (3), and $\mathbf{f}_{\mathbf{P}_{\mathbf{r},i}^t}$ is the estimated 2D backward optical flow at pixel $\mathbf{P}_{\mathbf{r},i}^t$.

**Total Variation Regularization.** We additionally employ a total variation prior [30] when training the 4D voxel deformation field to enforce the motion smoothness between neighboring voxels. At time step $t$,

$$\mathcal{L}_{\text{TV}}(t) = \frac{1}{2\bar{N}} \sum_{i=1}^{\bar{N}} \sum_{d \in C} \left( \Delta_x^2 (\mathbf{v}_i(t), d) + \Delta_y^2 (\mathbf{v}_i(t), d) + \Delta_z^2 (\mathbf{v}_i(t), d) \right) , \tag{7}$$

where $\Delta_{x,y,z}^2$ is the squared difference of motion vectors between voxel $\mathbf{v}_i$ and its neighbors along $x, y, z$ axes. $\bar{N} = N_x \times N_y \times N_z$ denotes the number of voxels.

**Training Objective.** The overall training objective of DeVRF is the combination of per-pixel re-rendering loss $\mathcal{L}_{\text{Render}}$, cycle consistency loss $\mathcal{L}_{\text{Cycle}}$, optical flow loss $\mathcal{L}_{\text{Flow}}$, and total variation regularization $\mathcal{L}_{\text{TV}}$:

$$\mathcal{L} = \omega_{\text{Render}} \cdot \mathcal{L}_{\text{Render}} + \omega_{\text{Cycle}} \cdot \mathcal{L}_{\text{Cycle}} + \omega_{\text{Flow}} \cdot \mathcal{L}_{\text{Flow}} + \omega_{\text{TV}} \mathcal{L}_{\text{TV}} , \tag{8}$$

where $\omega_{\text{Render}}, \omega_{\text{Cycle}}, \omega_{\text{Flow}}, \omega_{\text{TV}}$ are weights for corresponding losses.

Table 2: Averaged quantitative evaluation on inward-facing synthetic and real-world scenes against baselines and ablations of our method. We color code each cell as `best`, `second best`, and `third best`.

| | SYNTHETIC INWARD-FACING | | | | | REAL-WORLD INWARD-FACING | | | | |
|---|---|---|---|---|---|---|---|---|---|---|
| | PSNR↑ | SSIM↑ | LPIPS↓ | GPU (GB)↓ | Time↓ | PSNR↑ | SSIM↑ | LPIPS↓ | GPU (GB)↓ | Time↓ |
| Neural Volumes [18] | 9.620 | 0.532 | 0.5520 | 19.4 | 22.4hrs | 17.29 | 0.608 | 0.3440 | 19.2 | 22.0hrs |
| D-NeRF [27] | 31.83 | 0.960 | 0.0355 | 10.0 | 18.4hrs | 29.15 | 0.946 | 0.0643 | 12.4 | 22.1hrs |
| D-NeRF [27]-2 stage | 28.29 | 0.945 | 0.0528 | 9.7 | 18.4hrs | 27.21 | 0.936 | 0.0706 | 13.2 | 22.2hrs |
| D-NeRF [27]-dynamic | 17.59 | 0.839 | 0.2058 | 9.8 | 21.9hrs | 21.74 | 0.911 | 0.0906 | 13.5 | 22.3hrs |
| Nerfies [24] | 33.09 | 0.989 | 0.0432 | 21.8 | 18.7hrs | 29.58 | 0.980 | 0.0576 | 22.5 | 19.1hrs |
| Nerfies [24]-2 stage | 32.37 | 0.991 | 0.0322 | 22.0 | 15.8hrs | 23.93 | 0.920 | 0.0878 | 22.0 | 19.7hrs |
| Nerfies [24]-dynamic | 19.45 | 0.794 | 0.1674 | 22.0 | 21.3hrs | 20.70 | 0.910 | 0.1080 | 22.0 | 19.6hrs |
| HyperNeRF [25] | 33.73 | 0.965 | 0.0335 | 22.5 | 20.5hrs | 28.50 | 0.944 | 0.0692 | 22.0 | 20.5hrs |
| HyperNeRF [25]-2 stage | 29.16 | 0.953 | 0.0555 | 22.5 | 19.2hrs | 26.53 | 0.935 | 0.0802 | 22.0 | 19.3hrs |
| HyperNeRF [25]-dynamic | 18.00 | 0.786 | 0.2173 | 22.4 | 20.6hrs | 10.39 | 0.734 | 0.3990 | 22.0 | 20.5hrs |
| NSFF [16] | 27.06 | 0.936 | 0.0800 | 21.4 | 12.8hrs | 28.44 | 0.939 | 0.0714 | 22.7 | 15.3hrs |
| NSFF [16]-dynamic | 18.18 | 0.858 | 0.1929 | 15.0 | 15.5hrs | 19.90 | 0.909 | 0.0944 | 22.7 | 16.2hrs |
| Ours (base) | 22.44 | 0.887 | 0.1173 | 4.4 | 8mins | 24.56 | 0.917 | 0.0844 | 6.6 | 10mins |
| Ours w/ c2f | 31.97 | 0.975 | 0.0185 | 4.4 | 7mins | 27.83 | 0.956 | 0.0465 | 6.6 | 10mins |
| Ours w/ c2f, tv | 32.73 | 0.963 | 0.0172 | 4.4 | 7mins | 29.35 | 0.959 | 0.0434 | 6.6 | 10mins |
| Ours w/ c2f, tv, cycle | 33.97 | 0.981 | 0.0142 | 4.4 | 8mins | 31.56 | 0.971 | 0.0292 | 6.6 | 11mins |
| Ours w/ c2f, tv, cycle, flow | 34.29 | 0.982 | 0.0137 | 4.4 | 8mins | 31.68 | 0.972 | 0.0289 | 6.6 | 11mins |

# 4 Experiments

We extensively evaluate the DeVRF on various types of datasets, including five synthetic[3] 360° inward-facing dynamic scenes, two real-world forward-facing dynamic scenes, and one real-world 360° inward-facing dynamic scene. We run all experiments on a single NVIDIA GeForce RTX3090 GPU. During training, we set $\omega_{\text{Render}} = 1$, $\omega_{\text{Cycle}} = 100$, $\omega_{\text{Flow}} = 0.005$, and $\omega_{\text{TV}} = 1$ for all scenes.

## 4.1 Comparisons with SOTA Approaches

To demonstrate the performance of DeVRF, we compare DeVRF to various types of SOTA approaches, including a volumetric method Neural Volumes [18], NeRF-based methods D-NeRF [27], Nerfies [24], HyperNeRF [25], and a time-modulated method NSFF [16]. For a fair comparison, since DeVRF follows a static → dynamic learning paradigm, we additionally implement 2-stage versions of D-NeRF, Nerfies, and HyperNeRF to learn a canonical space prior in the first stage and then optimize a deformation network in the second stage. To show the effectiveness of our low-cost capture strategy for dynamic scenes, we also train these baselines using only a few-view dynamic sequences and observe a significant performance drop compared to those trained with both static and dynamic data. For quantitative comparison, peak signal-to-noise ratio (PSNR), structural similarity index (SSIM) [40], and Learned Perceptual Image Patch Similarity (LPIPS) [47] with VGG [33] are employed as evaluation metrics [4].

**Evaluation on inward-facing synthetic and real-world deformable scenes.** We selected five synthetic dynamic scenes with various types of deformations and motions, and rendered synthetic images in $400 \times 400$ pixels under the 360° inward-facing setup. For each scene, we use 100-view static images and 4-view dynamic sequences with 50 frames (*i.e.*, time steps) as training data for all approaches, and randomly select another 2 views at each time step for test. In addition, we collected one 360° inward-facing real-world deformable scene in $540 \times 960$ pixels. With our data capture setup, only 4 cameras are required to capture dynamic scenes, and we choose 3 views of them as training data and the other view as test data.

We report the metrics of the real-world scene as well as the average metrics of five synthetic scenes for all approaches in Tab. 2 and leave the per-scene metrics to supplementary material. As shown in Tab. 2, for synthetic and real-world scenes, DeVRF achieves the best performance in terms of PSNR and LPIPS, and the second- or third-best in terms of SSIM among all approaches. Most importantly, our per-scene optimization only takes less than 10mins with 4.4GB to 6.6GB GPU memory on a

---

[3]The *Lego* scene is shared by NeRF [20], licensed under the Creative Commons Attribution 3.0 License: https://creativecommons.org/licenses/by/3.0/. Other scenes are purchased from TurboSquid, licensed under the TurboSquid 3D Model License: https://blog.turbosquid.com/turbosquid-3d-model-license/.

[4]Although improving test-time rendering speed is not the focus of our paper, DeVRF achieves $16\times \sim 32\times$ test-time rendering speedup compared with other approaches, averaging 0.875 seconds per $540 \times 960$ image.

For the red box region of each scene, we show its zoom-in at the bottom-right of each picture

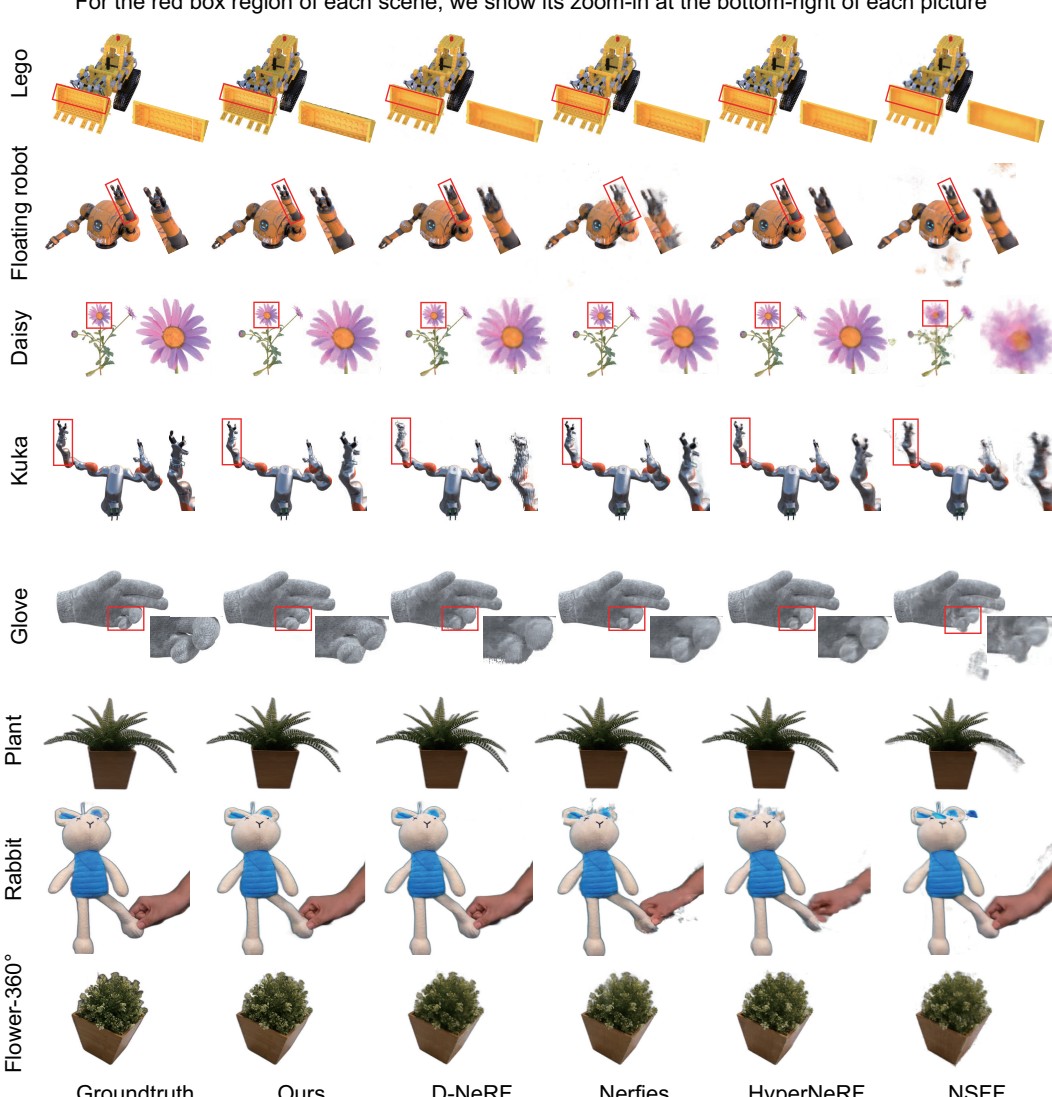

Figure 3: Qualitative comparisons of baselines and DeVRF on synthetic and real-world scenes.

single NVIDIA GeForce RTX3090 GPU, which is about two orders of magnitude faster than other approaches. The above quantitative comparison demonstrates the efficiency and effectiveness of DeVRF. Besides, the qualitative results of DeVRF and baselines on synthetic and real-world scenes are illustrated in Fig. 3, where DeVRF achieves on-par high-fidelity in comparison to SOTA methods. Please see the supplementary video for more results.

For a fair comparison, we additionally report the results of the 2-stage versions for D-NeRF [27], Nerfies [24], and HyperNeRF [25] in Tab. 2. Since these approaches are not designed to separately learn a canonical space and a deformation field, there is no significant difference in the results between their 2-stage versions and 1-stage versions using the training dataset. Furthermore, we also report the results of these baselines trained with dynamic data only (baseline-dynamic) in Tab. 2. Their performances drop significantly compared to the results trained with both static and dynamic data. In addition, since Neural Volumes [18] requires dozens of dynamic sequences as input, its performance is poor with our few-view dynamic sequences. The experimental results not only show the effectiveness of our low-cost data capture process and the proposed DeVRF model; but also validate our observation that few-view dynamic sequences alone fail to provide complete information about the dynamic scene, while static multi-view data can favorably serve as a supplement.

Table 3: Quantitative evaluation on forward-facing real-world scenes against baselines and ablations of our system. We color code each cell as best , second best , and third best .

| | PLANT | | | | | RABBIT | | | | |
|---|---|---|---|---|---|---|---|---|---|---|
| | PSNR↑ | SSIM↑ | LPIPS↓ | GPU (GB)↓ | Time↓ | PSNR↑ | SSIM↑ | LPIPS↓ | GPU (GB)↓ | Time↓ |
| D-NeRF [27] | 31.94 | 0.979 | 0.0251 | 11.4 | 21.5hrs | 33.51 | 0.974 | 0.0384 | 11.4 | 21.7hrs |
| Nerfies [24] | 31.36 | 0.991 | 0.0309 | 21.9 | 18.3hrs | 24.83 | 0.952 | 0.0795 | 21.9 | 19.8hrs |
| HyperNeRF [25] | 32.08 | 0.978 | 0.0331 | 22.0 | 20.0hrs | 24.97 | 0.942 | 0.0849 | 22.0 | 20.7hrs |
| NSFF [16] | 29.45 | 0.966 | 0.0526 | 20.2 | 14.5hrs | 27.68 | 0.945 | 0.0854 | 20.2 | 14.5hrs |
| Ours (base) | 26.13 | 0.946 | 0.0722 | 8.1 | 8mins | 25.58 | 0.910 | 0.1300 | 8.9 | 6mins |
| Ours w/ c2f | 31.85 | 0.980 | 0.0275 | 8.1 | 8mins | 26.79 | 0.938 | 0.0946 | 8.9 | 6mins |
| Ours w/ c2f, tv | 31.89 | 0.980 | 0.0263 | 8.1 | 8mins | 29.28 | 0.951 | 0.0655 | 8.9 | 6mins |
| Ours w/ c2f, tv, cycle | 31.99 | 0.981 | 0.0235 | 8.1 | 9mins | 31.05 | 0.963 | 0.0543 | 8.9 | 7mins |
| Ours w/ c2f, tv, cycle, flow | 32.01 | 0.981 | 0.0236 | 8.1 | 10mins | 32.05 | 0.966 | 0.0492 | 8.9 | 7mins |

**Evaluation on forward-facing real-world deformable scenes.** We collected two forward-facing real-world deformable scenes in $540 \times 960$ pixels using 4 cameras, and we chose 3 views of them as training data and the other view as test data. To handle forward-facing scenes, we adapt DeVRF to use normalized device coordinates (NDC) and multi-plane images (MPI) as in DVGO [35]. As shown in Tab. 3, DeVRF achieves the best result in the *plant* scene in terms of LPIPS metric and the second-best result in the *rabbit* scene in terms of all metrics. Fig. 3 also demonstrates qualitative comparisons on these two scenes.

## 4.2 Ablation Study

**Ablation study of DeVRF components.** We carry out ablation studies on both synthetic and real-world scenes to evaluate the effectiveness of each proposed component in DeVRF. We progressively ablate each component from optical flow, cycle consistency, total variation, to coarse-to-fine strategy. As shown in Tab. 2 and 3, the performance of DeVRF progressively drops with the disabling of each component, where disabling the coarse-to-fine training strategy causes the most significant performance drop. This is as expected since the coarse-to-fine training strategy is critical to reducing local minimums during optimization.

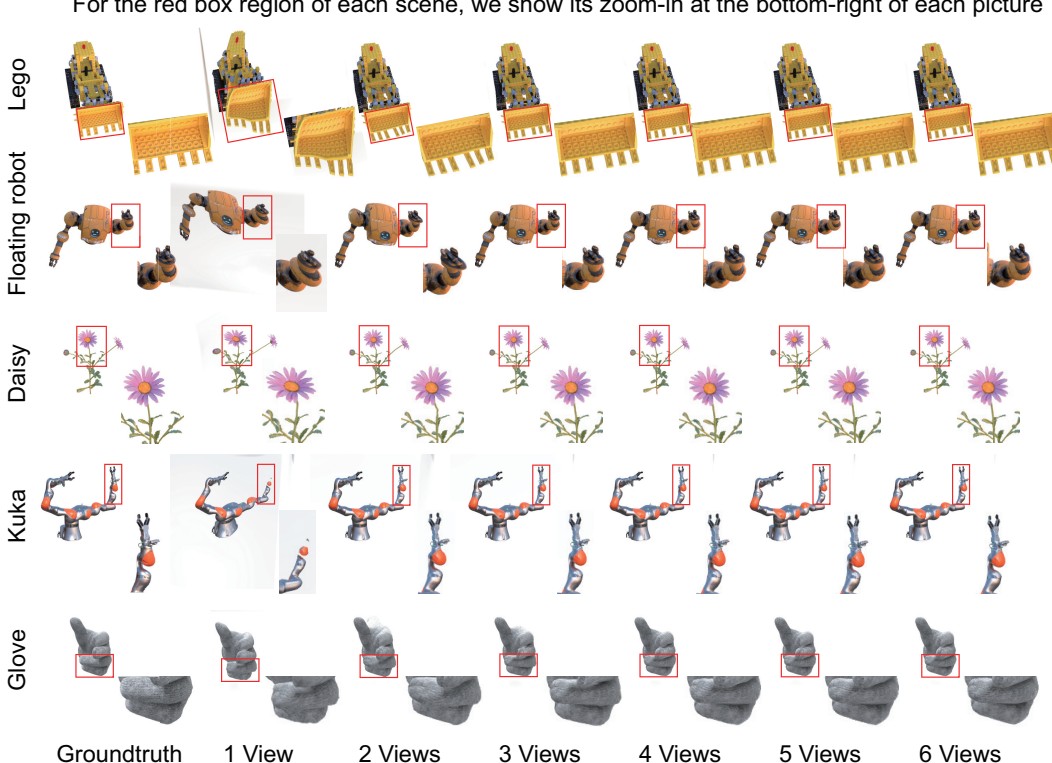

Figure 4: Qualitative results of DeVRF with different numbers of dynamic training views.

**Ablation study of dynamic training views on synthetic dataset.** Our static → dynamic learning paradigm is based on a low-cost yet effective capture setup: the multi-view static images provide complete 3D geometry and appearance information of the scene, while few-view dynamic sequences show how the scene deforms in 3D space over time. The entire capture process only requires a few cameras that are convenient to deploy in practice. To evaluate the influence of the number of dynamic views, we conduct additional ablations for DeVRF on five inward-facing synthetic dynamic scenes and report the per-scene metrics as well as the average metrics with respect to the number of dynamic training views. As shown in Fig. 5, given the same multi-view static images, the performance of DeVRF largely improves with the increment of dynamic training views and almost saturates at six dynamic views, and the four dynamic training views used in our paper can yield comparable results compared to six dynamic views. We additionally visualize the qualitative results of DeVRF with different numbers of dynamic training views in Fig. 4. Therefore, in our *static → dynamic* learning paradigm, with static multi-view data as a supplement, only a few (e.g., four) dynamic views are required to significantly boost the performance of dynamic neural radiance fields reconstruction. This further demonstrates the effectiveness of our low-cost data capture process and the DeVRF model.

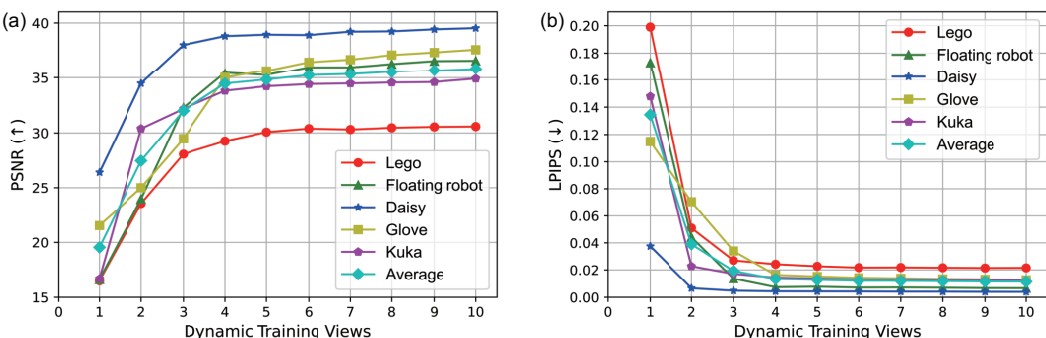

Figure 5: Ablation evaluation on the number of dynamic training views: (a) PSNR, (b) LPIPS.

# 5   Conclusion

We introduced DeVRF, a novel approach to tackle the challenging task of fast non-rigid radiance field reconstruction by modeling both the 3D canonical space and 4D deformation field of a dynamic scene with voxel-based representations. The DeVRF can be efficiently optimized in two major steps. We first proposed a *static → dynamic* learning paradigm to pinpoint that the 3D volumetric canonical prior can be effectively transferred into the 4D voxel deformation field. Second, based on this learning paradigm, we developed a series of optimization strategies, including coarse-to-fine learning, deformation cycle consistency, optical flow supervisions, and total variation priors. Such DeVRF finally produced a 100× faster training efficiency with on-par high-fidelity results in comparison to SOTA approaches. We believe our DeVRF can provide a complement to existing literature and new insights into the view synthesis community.

**Limitations and Future Work.**  Although DeVRF achieves fast deformable radiance field reconstruction, the model size is large due to its large number of parameters. In addition, DeVRF currently does not synchronously optimize the 3D canonical space prior during the second stage, and thus may not be able to model drastic deformations. We consider these limitations as faithful future directions.

# 6   Acknowledgements

This project is supported by the National Research Foundation, Singapore under its NRFF Award NRF-NRFF13-2021-0008, Mike Zheng Shou's Start-Up Grant from NUS, and National Research Foundation, Singapore and A*STAR, under its RIE2020 Industry Alignment Fund – Industry Collaboration Projects (IAF-ICP) grant call (Grant No. I2001E0059). The computational work for this article was partially performed on resources of the National Supercomputing Centre, Singapore. Jia-Wei Liu is supported by NUS IDS-ISEP scholarship.

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
