# OpenReview forum: "DeVRF: Fast Deformable Voxel Radiance Fields for Dynamic Scenes"
_NeurIPS.cc/2022/Conference — NeurIPS 2022 Accept_

### Official Review · Reviewer_yW6h · 2022-07-08

**Rating:** 6
**Confidence:** 3
**Soundness:** 3 good
**Presentation:** 3 good
**Contribution:** 3 good

**Summary:**

Authors propose an approach for novel view synthesis and reconstruction of dynamic scenes using a NeRF like model. The main idea is to split the capture/training phase between static and dynamic. Objects are first captured in a static setting using a single moving camera to build a canonical space. The dynamic performance is handled relying on multiple views rather than a single camera. A motion field is then learned to handle the dynamic of the scenes. Finally, temporal consistency enforces smooths results.

**Questions:**

I'd like authors to mostly comment on the weaknesses above. I think the paper has overall some interesting ideas that could be relevant for the community.

**Limitations:**

Some limitations are described, I would ask author to elaborate more on the actual capture requirements too. For instance, the dynamic part seems to need a multi-view setup, which limits the application in practice.

**Strengths And Weaknesses:**

++ Relevant problem, very active research topic in the community. Related work seems adequate.

++ Clarity. The paper is well described and reproducibility should not be a problem.

+/- Contributions: while the idea is definitely interesting, it is not clear the actual improvement over HyperNeRF (or even Nerfies). The results are marginally better and not clear why one should rely on this method instead of HyperNerf for instance. Nevertheless I find this idea of static and dynamic captures is interesting and I would suggest authors to add more results to support the claims. For instance, I would try the method on multi-view human datasets, where a single, multi-view frame could be used to learn the canonical space and then the rest of the sequence should learn a motion field. This could be particularly relevant for performance capture/avatars.

-- Experiments and Results. Authors should show more real-world examples. In particular results with less views for the dynamic part. I appreciated the quantitative results in the supplementary material, but would be good to show qualitatively the effect of using less views.

---

> ### Author Response · Authors · 2022-08-02
> **Response to Reviewer yW6h (2/2)**
>
> ### Q3: Results with less views for the dynamic part.
> Thank you for your comments. Following your suggestion, we additionally conduct an ablation study on the number of dynamic training views for 4 real-world scenes, and report the quantitative results (PSNR) in the following table. In addition, we have uploaded the test visualizations of different training views for 5 synthetic ([https://anonymous.4open.science/r/DeVRF-5520/visualization_results/vis_ablation_views_synthetic.pdf](https://anonymous.4open.science/r/DeVRF-5520/visualization_results/vis_ablation_views_synthetic.pdf)) and 4 real-world scenes ([https://anonymous.4open.science/r/DeVRF-5520/visualization_results/vis_ablation_views_real-world.pdf](https://anonymous.4open.science/r/DeVRF-5520/visualization_results/vis_ablation_views_real-world.pdf)).
>
> |             | 1 dynamic view | 2 dynamic views | 3 dynamic views |
> |-------------|----------------|-----------------|-----------------|
> | Pig         | 17.04          | 29.65           | 30.84           |
> | Plant       | 28.33          | 30.22           | 32.01           |
> | Rabbit      | 19.08          | 25.88           | 32.05           |
> | Flower-360° | 24.78          | 26.38           | 31.68           |
> *Tab. 2 Ablation evaluation on the number of dynamic training views for real-world scenes.*
>
>
> ### Q4: About actual capture requirements.
> We would like to clarify that our capture strategy is low-cost and easy to deploy in practice. We compared our capturing strategy with existing capturing approaches in Table 1 of the main paper. Existing methods either require dozens of high-quality cameras or are restricted to forward-facing scenes due to their setting of using a moving camera to capture deformable scenes. In comparison, we believe our capturing stage has the following benefits:
> 1.  Our capture strategy only requires 4 cameras and is very low-cost compared to existing multi-camera rigs that require dozens of high-quality cameras.
> 2.  Our capture device is easy and convenient to deploy in practice. In the first stage, capturing a static scene with a moving camera is easy and feasible. In addition, deploying a few (4) cameras requires much less human labor compared to building a dense rig with dozens of cameras.
> 3.  Our capture strategy can capture fast deformable scenes in both 360° inward-facing and forward facing settings. Existing approaches capture forward-facing deformable scenes with a moving camera. However, it would be very difficult to deploy this capturing strategy in 360° inward-facing fast deformable scenes, as the camera has to be moved as fast as the scene deforms to ensure decent view and motion coverage. For example, in our flower-360° scene, the flower vibrates with a very high frequency, and our capturing method can conveniently capture this kind of fast deformable scene at a very low cost. While existing approaches either require a dense camera rig (Fourier PlenOctrees) or cannot capture this scene (Nerfies).
>
> Therefore, we believe our capturing strategy **is easy to deploy in practice** with low money and labor cost, and thus provides a great complement to existing capturing strategy.
>
>
>
> [A] Peng, Sida, et al. "Neural body: Implicit neural representations with structured latent codes for novel view synthesis of dynamic humans." *Proceedings of the IEEE/CVF Conference on Computer Vision and Pattern Recognition*. 2021.

---

> > ### Comment · Reviewer_yW6h · 2022-08-08
> > **Thanks!**
> >
> > Thank you for the additional details. I think these clarify all my concerns and I am still in favor of acceptance.

---

> ### Author Response · Authors · 2022-08-02
> **Response to Reviewer yW6h (1/2)**
>
> Thank you for recognizing our work and the valuable comments.
>
> ### Q1: Improvements over HyperNeRF (and other baselines).
> Firstly, we would like to clarify that the main contribution of our DeVRF model is 100× faster convergence speed (10 minutes training on a single NVIDIA GeForce RTX3090 GPU) compared to previous SOTA approaches, with on-par high-fidelity rendering results. The core of the fast convergence of DeVRF is to model both the 3D canonical space and 4D deformation field of a dynamic, non-rigid scene with explicit and discrete voxel based representations. This explicit model representation of DeVRF enables fast querying of the deformation, density, color of a dynamic scene via linear interpolation. Together with our designed training strategies, the DeVRF can be optimized 100x faster than SOTA approaches with on-par high-fidelity results.
>
> Secondly, our DeVRF yields better rendering results on the local details of synthetic scenes and real-world scenes compared to previous approaches, such as the Lego shovel, Robot and Kuka fingers, Glove textures, Rabbit, Plant and Flower. The quality improvements of DeVRF are more obvious on real-world scenes, and we have updated the synthetic visualizations in Fig. 3 of the main paper in the anonymous link ([https://anonymous.4open.science/r/DeVRF-5520/visualization_results/vis_highlights_synthetic.pdf](https://anonymous.4open.science/r/DeVRF-5520/visualization_results/vis_highlights_synthetic.pdf)) to highlight the differences.
>
>
> ### Q2: Results on multi-view human capture datasets.
> Thank you for raising this great suggestion. We conduct additional experiments on a human body dataset that has highly variable poses - ZJU-MoCap dataset. We use both static multi-view images and 4 dynamic training views for training, and compare our DeVRF with D-NeRF. As shown in Tab. 2, our method achieves better performance with 100× faster convergence compared with D-NeRF. We find Nerfies and HyperNeRF prone to local minimums when trained on the challenging human body sequence and do not yield meaningful results, thus we do not report their results here. Note that we do not use any human shape or pose priors, thus we do not include comparisons with the line of work that focuses on human bodies (e.g., NeuralBody [A]). We have uploaded qualitative results on the anonymous link ([https://anonymous.4open.science/r/DeVRF-5520/visualization_results/vis_pig_toy.pdf](https://anonymous.4open.science/r/DeVRF-5520/visualization_results/vis_pig_toy.pdf) and [https://anonymous.4open.science/r/DeVRF-5520/visualization_results/vis_ZJU-MoCap.pdf](https://anonymous.4open.science/r/DeVRF-5520/visualization_results/vis_ZJU-MoCap.pdf)). Integrating the 3D human pose prior into our current DeVRF model would be a promising future direction. We will include experimental results on all other sequences of ZJU-MoCap in the revision.
>
> |        | PSNR ↑ | SSIM ↑ | LPIPS ↓ | GPU (GB) ↓ | Time ↓ |
> |--------|--------|--------|---------|------------|--------|
> | DeVRF  | 27.29  | 0.931  | 0.0210  | 5          | 8mins  |
> | D-NeRF | 19.76  | 0.811  | 0.1622  | 11         | 20hrs  |
> *Tab. 1 Quantitative comparisons on the ZJU-MoCap dataset ([Sequence CoreView_377]).*

---

### Official Review · Reviewer_7cQ7 · 2022-07-09

**Rating:** 4
**Confidence:** 4
**Soundness:** 2 fair
**Presentation:** 3 good
**Contribution:** 2 fair

**Summary:**

Modeling geometry and motions for a dynamic scene is an important task in computer vision. This paper proposes a volumetric representation to speed up learning dynamic radiance fields. It employs a 4D voxel grid to model deformations while continuing to use a 3D voxel grid to model the canonical space. To save optimization time and provide well initialization, this paper devises a 2-stage capturing procedure – the 1st stage is to capture a static state with a moving monocular camera, and the 2nd stage is to capture dynamic states with a few fixed cameras. Accordingly, the captured images in the 1st stage are utilized for training the canonical space, while the captured images in the 2nd stage are utilized for training the deformation field.

**Questions:**

1, The design of the backward deformation is well-written, but the process of obtaining the forward motion is not clearly presented. Could you explain how to query the corresponding points of a canonical 3D point at different time steps?
2, Could you verify that the proposed DeVRF can still receive a well rendering performance without the 2-stages training strategy, like evaluating DeVRF with optical flow, cycle consistency, total variation, as well as a general training process? And compare it with the mainstream frameworks (D-NeRF, Nerfies and HyperNeRF).
3, If the users are able to employ a few (4) fixed cameras in capturing dynamic states, why do not use the binocular camera for recovering depth and then leverage a fusion strategy to model the deformation (a real-time procedure)?
4, Could you verify DeVRF in solving some challenging cases, like fast motions or highly variable poses?

**Limitations:**

The authors have discussed the limitations in the end.

**Strengths And Weaknesses:**

Strengths:
1.	The paper is well-written, including clarifying the motivation and introducing the experiment setup.
2.	The diagrammatic drawing (Fig. 1 and Fig. 2) is technical and clearly describes the claimed contributions.

Weakness
1.	The setup of capturing strategy is complicated and is not easy for applications in real life. To initialize the canonical space, the first stage is to capture the static state using a moving camera. Then to model motions, the second stage is to capture dynamic states using a few (4) fixed cameras. Such a 2-stage capturing is not straightforward.
2.	To utilize a volumetric representation in the deformation field is not a novel idea. In the real-time dynamic reconstruction task, VolumeDeform [1] has proposed volumetric grids to encode both the geometry and motion, respectively.
3.	The quantitative experiments (Tab. 2 and Tab. 3) show that the fidelity of rendered results highly depends on the 2-stage training strategy. In a general capturing case, other methods can obtain more accurate rendered images. Oppositely, Tab. 2 shows that it is not easy to fuse the designed 2-stage training strategy into current mainstream frameworks, such as D-NeRF, Nerfies and HyperNeRF. It verifies that the 2-stages training strategy is not a general design for dynamic NeRF.

[1] Innmann, Matthias, Michael Zollhöfer, Matthias Nießner, Christian Theobalt, and Marc Stamminger. "Volumedeform: Real-time volumetric non-rigid reconstruction." In European conference on computer vision (ECCV), pp. 362-379. Springer, Cham, 2016.

---

> ### Author Response · Authors · 2022-08-02
> **Response to Reviewer 7cQ7 (4/4)**
>
> ### Q6: Comparisons with depth estimation and fusion methods.
> Thanks for your question. Using multi-view stereo to estimate depth requires the camera views to have sufficient overlap, which cannot be guaranteed in the 360° inward-facing capture setting, where a few (4) cameras need to cover the entire scene. Wide baseline stereo methods might help, but we believe it is out of scope of this work. We do use closely placed cameras in forward-facing capture; however, it is still hard to recover accurate depth for textureless regions, areas presenting view-dependent effects, and thin structures (e.g., the human arm in our Rabbit scene, leaves in the Plant and Flower scenes). In contrast, our method can handle both the inward-facing 360° scenes and forward-facing scenes with 4 cameras, which is superior to depth-based approaches. Incorporating depth inputs might be a promising direction, and we would like to leave it as future work.
>
>
> ### Q7: More results on challenging datasets.
> Our model can handle fast motions. As shown in our supplementary video, the plant and flower scenes vibrate in a high frequency and our model can predict high-quality rendering results in these scenes. For scenes with highly variable poses, we conduct additional experiments on a sequence of the human body dataset - ZJU-MoCap. We use both static multi-view images and 4 dynamic training views for training, and compare our DeVRF with D-NeRF. As shown in Tab. 2, our method achieves better performance with 100× faster convergence compared with D-NeRF. We find Nerfies and HyperNeRF prone to local minimums when trained on the challenging human body sequence and do not yield meaningful results, thus we do not report their results here. Note that we do not use any human shape or pose priors, thus we do not include comparisons with the line of work that focuses on human bodies (e.g., NeuralBody [A]). We have uploaded qualitative results on the Anonymous Link ([https://anonymous.4open.science/r/DeVRF-5520/visualization_results/vis_pig_toy.pdf](https://anonymous.4open.science/r/DeVRF-5520/visualization_results/vis_pig_toy.pdf) and [https://anonymous.4open.science/r/DeVRF-5520/visualization_results/vis_ZJU-MoCap.pdf](https://anonymous.4open.science/r/DeVRF-5520/visualization_results/vis_ZJU-MoCap.pdf)). Integrating the 3D human pose prior into our current DeVRF model would be a promising future direction. We will include experimental results on all other sequences of ZJU-MoCap in the revision.
>
> |        | PSNR ↑ | SSIM ↑ | LPIPS ↓ | GPU (GB) ↓ | Time ↓ |
> |--------|--------|--------|---------|------------|--------|
> | DeVRF  | 27.29  | 0.931  | 0.0210  | 5          | 8mins  |
> | D-NeRF | 19.76  | 0.811  | 0.1622  | 11         | 20hrs  |
> *Tab. 2 Quantitative comparisons on the ZJU-MoCap dataset ([Sequence CoreView_377]).*
>
>
> [A] Peng, Sida, et al. "Neural body: Implicit neural representations with structured latent codes for novel view synthesis of dynamic humans." *Proceedings of the IEEE/CVF Conference on Computer Vision and Pattern Recognition*. 2021.

---

> ### Author Response · Authors · 2022-08-02
> **Response to Reviewer 7cQ7 (3/4)**
>
> ### Q4: About forward motion.
> The forward deformation is modeled as 4D forward deformation voxels with the same resolution as the backward deformation. The 3D forward motion $\Delta\mathcal{X}_{0\rightarrow t}=\{\Delta\mathbf{X}_i^{0\rightarrow t}\}$ from canonical 3D points $\mathcal{X}_0$ to their corresponding 3D points at time step $t$, $\mathcal{X}_t = \{\mathbf{X}_i^{t}~\rvert~\mathbf{X}_i^{t} = \mathbf{X}_i^{0} + \Delta\mathbf{X}_i^{0\rightarrow t}\}$ (please see the [link](https://anonymous.4open.science/r/DeVRF-5520/visualization_results/eq_upload.pdf) for a properly compiled equation) can be efficiently queried through quadruple interpolation of their neighboring voxels at neighboring time steps in the 4D forward deformation field, as shown in Eq. (2) of the main paper. The 4D forward deformation field is jointly optimized with the 4D backward deformation field through 4D deformation cycle consistency and optical flow supervision.
>
>
> ### Q5: Could you verify that the proposed DeVRF can still receive a well rendering performance without the 2-stages training strategy, like evaluating DeVRF with optical flow, cycle consistency, total variation, as well as a general training process? And compare it with the mainstream frameworks (D-NeRF, Nerfies and HyperNeRF)
> Our DeVRF is designed to accelerate learning dynamic radiance fields in a novel static → dynamic capture setting. This setting is different from Nefies setting because our setting can integrate the static multi-view information with few-view dynamic sequences, thus enabling capturing both 360° inward-facing and forward-facing dynamic scenes at a low cost. We have evaluated the efficiency and effectiveness of DeVRF in various deformable scenes, where some scenes even have very fast deformations such as the vibrations of flowers. The proposed DeVRF achieves 100× faster convergence than previous SOTA approaches with on-par high-fidelity results, and our approach uses about 2-5 times lower GPU memory than SOTA approaches. Extending DeVRF to other settings is not the main focus of our paper, but we believe our DeVRF model and the training strategy designs can be used in other settings. For example, D-NeRF adopts a sampling bias to mainly sample quasi-static images at the beginning of training, which can be conveniently adopted in DeVRF to extend DeVRF to one-stage training.

---

> ### Author Response · Authors · 2022-08-02
> **Response to Reviewer 7cQ7 (2/4)**
>
> ### Q3: The quantitative experiments (Tab. 2 and Tab. 3) show that the fidelity of rendered results highly depends on the 2-stage training strategy. In a general capturing case, other methods can obtain more accurate rendered images. Oppositely, Tab. 2 shows that it is not easy to fuse the designed 2-stage training strategy into current mainstream frameworks, such as D-NeRF, Nerfies and HyperNeRF. It verifies that the 2-stages training strategy is not a general design for dynamic NeRF.
> Thank you for the detailed discussion about our experiments, but there might be some misunderstandings in your comments, and we will answer your questions with our experimental results point by point.
>
> Firstly, experimental results in Tabs. 2 and 3 do not suggest the statement - “the fidelity of rendered results highly depends on the 2-stage training strategy”. For a fair comparison, we report the quantitative results of 3 versions of baselines in Tabs. 2 and 3: 1-stage version trained with both static and dynamic data (first row of each baseline), 2-stage version trained with both static and dynamic data (second row of each baseline, i.e., “-2 stage”), and 1-stage version trained with only dynamic data (third row of each baseline, i.e., “-dynamic”).  Since these baselines are not designed for 2-stage training, there is no significant difference in the results between their 2-stage versions and 1-stage versions using the same training dataset. In addition, the performances of baselines trained with dynamic data only drop significantly compared to the results trained with both static and dynamic data, which validates our observation that few-view dynamic sequences alone fail to provide complete information about the dynamic scene, while static multi-view data can favorably serve as a supplement.
>
> Secondly, experimental results in Tabs. 2 and 3 do not suggest the statement - “In a general capturing case, other methods can obtain more accurate rendered images”. We suppose that here “general capturing case” means using a moving camera to capture a deformable scene, as in D-NeRF and Nerfies. To further compare our capture strategy with the “general capturing case”, we additionally collect 5 360° inward-facing synthetic scenes captured with a 360° rotating camera, where the camera is positioned at sequential poses at different timesteps (50 timesteps in our experiments). While in our 2-stage setting, we only use 4 fixed camera poses to capture a deformable scene during the second stage, and complement the few-view dynamic sequences with multi-view static images captured in the first stage. We additionally train D-NeRF, Nerfies, and HyperNeRF on our capture setting and the “general capturing case”, and we evaluate these baselines using the same test set. We report the averaged quantitative results of 5 synthetic scenes in the following Tab. 1, where baseline - ours means the baseline is trained with our capture setting, and baseline - general means the baseline is trained with the “general setting”. As shown in this table, the performance of baselines drops significantly when trained with the “general capturing case”. This clearly demonstrates the effectiveness of our 2-stage static-dynamic capture strategy.
>
> |                     | PSNR ↑ | SSIM ↑ | LPIPS ↓ |
> |---------------------|--------|--------|---------|
> | D-NeRF - ours       | 31.83  | 0.960  | 0.0355  |
> | D-NeRF - general    | 22.66  | 0.896  | 0.1073  |
> | Nerfies - ours      | 33.09  | 0.989  | 0.0432  |
> | Nerfies - general   | 18.75  | 0.791  | 0.1473  |
> | HyperNeRF - ours    | 33.73  | 0.965  | 0.0335  |
> | HyperNeRF - general | 21.48  | 0.895  | 0.1300  |
> *Tab. 1 Quantitative comparisons between our 2-stage capture setting and general setting*
>
> Thirdly, experimental results in Tab. 2 do not suggest the statement - “Tab. 2 shows that it is not easy to fuse the designed 2-stage training strategy into current mainstream frameworks, such as D-NeRF, Nerfies and HyperNeRF. It verifies that the 2-stages training strategy is not a general design for dynamic NeRF.”. In our implementation, it is actually quite easy to fuse our designed 2-stage training strategy to these baselines. Since these baselines are not designed for 2-stage training, there is no significant difference in the results between their 2-stage versions and 1-stage versions using the same training dataset. Notably, as shown in Tab. 2 in the main paper, the 2-stage version of Nerfies achieves better results than its 1-stage version in terms of SSIM and LPIPS on 5 synthetic scenes.

---

> ### Author Response · Authors · 2022-08-02
> **Response to Reviewer 7cQ7 (1/4)**
>
> Thank you for your helpful comments.
>
> ### Q1: About the capture setup.
> We compared our capturing strategy with existing capturing strategies in Table 1 of the main paper. Existing methods either require dozens of high-quality cameras or are restricted to forward-facing scenes due to their setting of using a moving camera to capture deformable scenes. In comparison, we believe our capturing stage has the following benefits:
> 1.  Our capture strategy only requires 4 cameras and is very low-cost compared to existing multi-camera rigs that require dozens of high-quality cameras.
> 2.  Our capture device is easy and convenient to deploy in practice. In the first stage, capturing a static scene with a moving camera is easy and feasible. In addition, deploying a few (4) cameras requires much less human labor compared to building a dense rig with dozens of cameras.
> 3.  Our capture strategy can capture fast deformable scenes in both 360° inward-facing and forward facing settings. Existing approaches capture forward-facing deformable scenes with a moving camera. However, it would be very difficult to deploy this capturing strategy in 360° inward-facing fast deformable scenes, as the camera has to be moved as fast as the scene deforms to ensure decent view and motion coverage. For example, in our flower-360° scene, the flower vibrates with a very high frequency, and our capturing method can conveniently capture this kind of fast deformable scene at low cost. While existing approaches either require a dense camera rig (e.g., Fourier PlenOctrees) or cannot capture this scene (e.g., Nerfies).
>
> Therefore, we believe our capturing strategy **is easy to deploy in practice** with low money and labor cost, and thus provides a great complement to existing capturing strategy.
>
>
> ### Q2: To utilize a volumetric representation in the deformation field is not a novel idea. In the real-time dynamic reconstruction task, VolumeDeform has proposed volumetric grids to encode both the geometry and motion, respectively.
> Thanks for your comment. We addressed VolumeDeform in the Related Work section in our main paper (Line 78-80) and will further clarify its contribution in introducing a unified volumetric representation to encode both the scene’s geometry and its motion. Nevertheless, to the best of our knowledge, we are the first to incorporate the 4D voxel deformation field into **dynamic radiance fields** (Line 64-65), and we design a set of strategies to fast optimize both the 4D voxel deformation field and 3D voxel canonical space in the dynamic radiance fields, thereby enabling fast high-fidelity dynamic novel view synthesis **from RGB inputs**. In contrast, VolumeDeform takes **RGB-D** images as input, where the depth can assist correspondence association and energy minimization. Besides, VolumeDeform mainly focuses on shape reconstruction, while DeVRF focuses on dynamic radiance field reconstruction for novel view synthesis.

---

### Official Review · Reviewer_7Cm1 · 2022-07-12

**Rating:** 6
**Confidence:** 4
**Soundness:** 3 good
**Presentation:** 3 good
**Contribution:** 3 good

**Summary:**

The paper proposed a novel view synthesis method for dynamic scenes using a deformable voxel formation. They show that using this representation is hard to train as there are large number of parameters. Sp device a static to dynamic paradigm to train faster resulting in 100x speed over previous dynamic Radiance fields.

**Questions:**

why was such a small dataset being proposed? It is easy to capture a larger data from the current configuration.



**Limitations:**

Limitations have been well addressed. It would be great if the authors provided negative societal impact for dynamic radiance fields.

**Strengths And Weaknesses:**

Strengths
- The paper is well written and easy to follow with key contributions well formulated and experimented.
- The dataset being used can be helpful in quickly starting the research in the dynamic reconstruction paradigm as it is low cost.
- Learning Deformable voxel radiance field helps understand dynamic scenes accurately and faster.
- Using the Static to dynamic learning paradigm is helpful in general for dynamic scene understanding and the coarse to fine optimization is useful in dynamic scenes.
- The ablation study shows the advantage of using different components in the optimization like optical flow to the overall accuracy of dynamic reconstruction

Weaknesses:
- The dataset captured in the real world is too small for accurate evaluation. Since the method is dynamic different motions of the same objects can be added to the dataset to make more evaluations and would make the method stronger.
- Fig 3 shows comparison to different methods but the synthetic examples seems to be very similar to other baselines. It is hard to understand the advantage of the current framework over such examples. Seems like the method performs well on the real videos. May be showing more such examples will be helpful.
- The method is proposed to be 100x faster, this contribution is mainly because of sparser input data with good flow. Doing more evaluations on what number of inputs is actually sufficient and are needed is will be an interesting insight.

---

> ### Author Response · Authors · 2022-08-02
> **Response to Reviewer 7Cm1 (2/2)**
>
> ### Q2: Qualitative results on synthetic datasets seem to be very similar to other baselines; more real-world examples.
> Firstly, we would like to clarify that the main contribution of our DeVRF model is 100× faster convergence speed (10 minutes training on a single NVIDIA GeForce RTX3090 GPU) compared to previous SOTA approaches, with on-par high-fidelity rendering results. Secondly, our DeVRF yields better rendering results on the local details of synthetic scenes compared to previous approaches, such as the Lego shovel, Robot and Kuka fingers, Glove textures. We have updated Fig. 3 of the main paper in the anonymous link ([https://anonymous.4open.science/r/DeVRF-5520/visualization_results/vis_highlights_synthetic.pdf](https://anonymous.4open.science/r/DeVRF-5520/visualization_results/vis_highlights_synthetic.pdf)) to highlight the differences. Finally, please refer to the reply to Q1 for more real-world examples.
>
> ### Q3: The method is proposed to be 100x faster, this contribution is mainly because of sparser input data with good flow. Doing more evaluations on what number of inputs is actually sufficient and are needed will be an interesting insight.
> Firstly, we would like to clarify that the core of the 100x faster convergence of DeVRF is not because of sparse input data. Instead, the core is to model both the 3D canonical space and 4D deformation field of a dynamic, non-rigid scene with explicit and discrete voxel based representations. This explicit model representation of DeVRF enables fast querying of the deformation, density, color of a dynamic scene via linear interpolation. Together with our designed training strategies, the DeVRF can be optimized 100x faster than SOTA approaches with on-par high-fidelity results.
>
> Secondly, in our supplementary materials (Section C - Additional Ablations), we have done additional ablation study on the number of dynamic training views on 5 synthetic scenes, and it shows that the performance of DeVRF improves with the increment of dynamic training views and almost saturates at six dynamic views, and the four dynamic training views used in our paper can yield comparable results compared to six dynamic views. Notably, we additionally conduct an ablation study on the number of dynamic training views for 4 real-world scenes and report their PSNR quantitative results in Tab. 3 below. In addition, we have uploaded the test visualizations of different training views for 5 synthetic scenes ([https://anonymous.4open.science/r/DeVRF-5520/visualization_results/vis_ablation_views_synthetic.pdf](https://anonymous.4open.science/r/DeVRF-5520/visualization_results/vis_ablation_views_synthetic.pdf)) and 4 real-world scenes ([https://anonymous.4open.science/r/DeVRF-5520/visualization_results/vis_ablation_views_real-world.pdf](https://anonymous.4open.science/r/DeVRF-5520/visualization_results/vis_ablation_views_real-world.pdf)).
>
> |             | 1 dynamic view | 2 dynamic views | 3 dynamic views |
> |-------------|----------------|-----------------|-----------------|
> | Pig         | 17.04          | 29.65           | 30.84           |
> | Plant       | 28.33          | 30.22           | 32.01           |
> | Rabbit      | 19.08          | 25.88           | 32.05           |
> | Flower-360° | 24.78          | 26.38           | 31.68           |
> *Tab. 3 Ablation evaluation on the number of dynamic training views for real-world scenes.*
>
> ### Limitation-1. Limitations have been well addressed. It would be great if the authors provided negative societal impact for dynamic radiance fields.
> We believe our work does not bring immediate negative societal impact. However, accurately reconstructed dynamic scenes may raise privacy concerns and thus need to be prevented from malevolent use.
>
>
>
> [A] Peng, Sida, et al. "Neural body: Implicit neural representations with structured latent codes for novel view synthesis of dynamic humans." *Proceedings of the IEEE/CVF Conference on Computer Vision and Pattern Recognition*. 2021.

---

> ### Author Response · Authors · 2022-08-02
> **Response to Reviewer 7Cm1 (1/2)**
>
> Thank you for recognizing our work and the valuable comments.
>
> ### Q1: Small real-world dataset.
> > Weakness 1: The dataset captured in the real world is too small for accurate evaluation; adding different motions of the same objects to the dataset.
>
> > Question 1: Why was such a small dataset being proposed? It is easy to capture a larger data from the current configuration.
>
> Firstly, we have added one more real-world scene where we squeeze and spin a pig toy. As shown in Tab. 1, our DeVRF achieves 100× convergence speed with on-par high-fidelity rendering results compared to previous SOTA approaches once again. Secondly, the ZJU-MoCap dataset fits the requirement of different motions of the same objects. Therefore, we have conducted additional experiments on a human body dynamic sequence of the ZJU-MoCap dataset. As shown in Tab. 2, our method achieves better performance with 100× faster convergence compared with D-NeRF. We find Nerfies and HyperNeRF prone to local minimums when trained on the challenging human body sequence and do not yield meaningful results, thus we do not report their results here. Note that we do not use any human shape or pose priors, thus we do not include comparisons with the line of work that focuses on human bodies (e.g., NeuralBody [A]). We report the quantitative results on the following tables, and upload qualitative results on the Anonymous Link ([https://anonymous.4open.science/r/DeVRF-5520/visualization_results/vis_pig_toy.pdf](https://anonymous.4open.science/r/DeVRF-5520/visualization_results/vis_pig_toy.pdf) and [https://anonymous.4open.science/r/DeVRF-5520/visualization_results/vis_ZJU-MoCap.pdf](https://anonymous.4open.science/r/DeVRF-5520/visualization_results/vis_ZJU-MoCap.pdf)). We will further include results on all other sequences of ZJU-MoCap in the revision. Finally, we would like to emphasize the importance of our 5 synthetic inward-facing scenes on evaluating our DeVRF. Since these synthetic datasets are rendered at 50 views during the second dynamics capturing stage, we can evaluate our DeVRF and baselines in the other 46 novel views.
>
> In summary, we further collected and conducted more real-world examples. We believe such a combination of synthetic and real-world scenes is adequate to verify the superiority of the proposed DeVRF: (1) 5 synthetic scenes (2) 4 real-world scenes with different types of objects, deformations, and camera arrangements (inward-facing or forward-facing), (3) the ZJU-Mocap dataset. Nonetheless, we will continue capturing more real-world scenes such as the same objects with different motions to enlarge our dataset and put the visualization results on our online project.
>
> |           | PSNR ↑ | SSIM ↑ | LPIPS ↓ | GPU (GB) ↓ | Time ↓ |
> |-----------|--------|--------|---------|------------|--------|
> | DeVRF     | 30.85  | 0.967  | 0.0447  | 7          | 10mins |
> | D-NeRF    | 31.23  | 0.974  | 0.0381  | 12         | 22hrs  |
> | Nerfies   | 30.66  | 0.986  | 0.0487  | 22         | 20hrs  |
> | HyperNeRF | 25.92  | 0.942  | 0.1037  | 22         | 21hrs  |
> *Tab. 1 Quantitative comparisons on the new real-world scene - Pig Toy.*
>
> |        | PSNR ↑ | SSIM ↑ | LPIPS ↓ | GPU (GB) ↓ | Time ↓ |
> |--------|--------|--------|---------|------------|--------|
> | DeVRF  | 27.29  | 0.931  | 0.0210  | 5          | 8mins  |
> | D-NeRF | 19.76  | 0.811  | 0.1622  | 11         | 20hrs  |
> *Tab. 2 Quantitative comparisons on the ZJU-MoCap dataset ([Sequence CoreView_377]).*

---

### Author Response · Authors · 2022-08-02
**Reply to AC and all reviewers (2/2)**

### 1. More experiments on challenging real-world scenes and human body datasets.
We have **added one more real-world scene** where we squeeze and spin a toy, and further evaluated our method on a sequence of the **challenging human body dataset ZJU-MoCap**, as shown in Tab. 1 and 2 below (qualitative results can be found at [https://anonymous.4open.science/r/DeVRF-5520/visualization_results/vis_pig_toy.pdf](https://anonymous.4open.science/r/DeVRF-5520/visualization_results/vis_pig_toy.pdf) and [https://anonymous.4open.science/r/DeVRF-5520/visualization_results/vis_ZJU-MoCap.pdf](https://anonymous.4open.science/r/DeVRF-5520/visualization_results/vis_ZJU-MoCap.pdf)). As expected, our DeVRF achieves 100× faster convergence speed with on-par high-fidelity rendering results compared to SOTA approaches in Tab.1. For the challenging human body motion sequence, our method achieves better performance with 100× faster convergence compared with D-NeRF. We find Nerfies and HyperNeRF prone to local minimums when trained on the challenging human body sequence and do not yield meaningful results, thus we do not report their results here. Note that we do not use any human shape or pose priors, thus we do not include comparisons with the line of work that focuses on human bodies (e.g., NeuralBody [A]). We will include experimental results on all other sequences of ZJU-MoCap in the revision.

|           | PSNR ↑ | SSIM ↑ | LPIPS ↓ | GPU (GB) ↓ | Time ↓ |
|-----------|--------|--------|---------|------------|--------|
| DeVRF     | 30.85  | 0.967  | 0.0447  | 7          | 10mins |
| D-NeRF    | 31.23  | 0.974  | 0.0381  | 12         | 22hrs  |
| Nerfies   | 30.66  | 0.986  | 0.0487  | 22         | 20hrs  |
| HyperNeRF | 25.92  | 0.942  | 0.1037  | 22         | 21hrs  |
*Tab. 1 Quantitative comparisons on the new real-world scene - Pig Toy.*


|        | PSNR ↑ | SSIM ↑ | LPIPS ↓ | GPU (GB) ↓ | Time ↓ |
|--------|--------|--------|---------|------------|--------|
| DeVRF  | 27.29  | 0.931  | 0.0210  | 5          | 8mins  |
| D-NeRF | 19.76  | 0.811  | 0.1622  | 11         | 20hrs  |
*Tab. 2 Quantitative comparisons on the ZJU-MoCap dataset ([Sequence CoreView_377]).*


### 2. Design of the 2-stage capturing setup.
We believe our capture setup is **low-cost** and **easy** to deploy in practice. In the first stage, we capture a static scene using a moving camera, and in the second stage, we capture a deformable scene using 4 fixed cameras. Our experiments demonstrate that the multi-view information is a great supplement for few-view dynamic sequences, and thus enables capturing dynamic scenes with only 4 cameras. This setup can capture both 360° inward-facing and forward-facing deformable scenes with different types of deformations, while existing methods heavily rely on dozens of high-quality cameras or are restricted to forward-facing scenes.

To further compare our capture strategy with the a general capturing setting that uses a moving camera to capture a deformable scene, we additionally collect 5 360° inward-facing synthetic scenes captured with a 360° rotating camera, where the camera is positioned at sequential poses at different timesteps (50 timesteps in our experiments). We additionally train D-NeRF, Nerfies, and HyperNeRF on our capture setting and the general capture setting, and we evaluate these baselines using the same test set. We report the averaged quantitative results of 5 synthetic scenes in the following Tab. 3, where baseline - ours means the baseline is trained with our capture setting, and baseline - general means the baseline is trained with the general setting. As shown in this table, the performance of baselines drops significantly when trained with the general capture setting. This clearly demonstrates the effectiveness of our 2-stage static-dynamic capture strategy.

|                     | PSNR ↑ | SSIM ↑ | LPIPS ↓ |
|---------------------|--------|--------|---------|
| D-NeRF - ours       | 31.83  | 0.960  | 0.0355  |
| D-NeRF - general    | 22.66  | 0.896  | 0.1073  |
| Nerfies - ours      | 33.09  | 0.989  | 0.0432  |
| Nerfies - general   | 18.75  | 0.791  | 0.1473  |
| HyperNeRF - ours    | 33.73  | 0.965  | 0.0335  |
| HyperNeRF - general | 21.48  | 0.895  | 0.1300  |
*Tab. 3 Quantitative comparisons between our 2-stage capture setting and general setting*

Besides, we have released the project (including model training detail, visualization results, etc.) in the Anonymous Link ([https://anonymous.4open.science/r/DeVRF-5520](https://anonymous.4open.science/r/DeVRF-5520)) to complement all the other possible missing details, and will **open-source** it after the double-blind reviewing.


[A] Peng, Sida, et al. "Neural body: Implicit neural representations with structured latent codes for novel view synthesis of dynamic humans." *Proceedings of the IEEE/CVF Conference on Computer Vision and Pattern Recognition*. 2021.

---

### Author Response · Authors · 2022-08-02
**Reply to AC and all reviewers (1/2)**

We thank all the reviewers for their insightful and valuable comments. We also appreciate that the core contributions and the quality of our results are recognized in the review:
1.  Our deformable voxel radiance field helps **understand dynamic scenes accurately and faster.** (Reviewer 7Cm1)
2.  The idea of static to dynamic learning paradigm is **definitely interesting**. (Reviewer yW6h)
3.  The paper is **well-written**, including **clarifying the motivation** and introducing the experiment setup. (Reviewer 7cQ7)

One common advice from all the reviewers is to include more results on real-world datasets. We have conducted additional experiments on the challenging ZJU-Mocap dataset and one more real-world captured scene, which further strengthen the contributions and effectiveness of the proposed method. Reviewers also have common concerns on the design and effectiveness of the 2-stage capturing setup. We have added quantitative comparisons of different capturing processes, which further support the rationale behind our 2-stage design. We will also improve the clarity of Section 3.1 according to the reviews in the revision.

Considering the technical contributions and promising results, we believe our submission can reach the bar of a NeurIPS paper after the revisions based on the advice from all reviewers. We would also like to mention that the dynamic neural radiance field has drawn considerable attention, and DeVRF may enlighten many follow-ups in the NeurIPS community.

Below, we respond to the two major concerns mentioned above in detail. For other concerns and comments, please refer to individual responses to each reviewer.

---

### Meta-Review · Area_Chair_fnQp · 2022-08-27

**Recommendation:** Accept
**Confidence:** Certain

**Metareview:**

Reviewers are in agreement that the paper addresses an important task (modelling dynamic scenes) and is well presented and written.

Reviewer 7cq7 considers that the 2-stage capture is a limitation, and it is correct that the most general solution, towards which the field strives to move, is a small number of moving or static cameras capturing a complex dynamic scene.  However, as noted by other reviewers, this configuration is not impractical, and study of this case will likely contribute to the field overall.


**Award:**

No

---

### Decision · Program_Chairs · 2022-09-14

Accept